# Host-Plant Selection Behavior of *Ophraella communa*, a Biocontrol Agent of the Invasive Common Ragweed *Ambrosia artemisiifolia*

**DOI:** 10.3390/insects14040334

**Published:** 2023-03-29

**Authors:** Jisu Jin, Meiting Zhao, Zhongshi Zhou, Ren Wang, Jianying Guo, Fanghao Wan

**Affiliations:** 1State Key Laboratory for Biology of Plant Diseases and Insect Pests, Institute of Plant Protection, Chinese Academy of Agricultural Sciences, Beijing 100193, China; jinjisu9110@163.com (J.J.);; 2School of Marxism, Ludong University, Yantai 264025, China; 3Agricultural Genomics Institute at Shenzhen, Chinese Academy of Agricultural Sciences, Shenzhen 518120, China; 4College of Plant Health and Medicine, Qingdao Agricultural University, Qingdao 266109, China

**Keywords:** host specificity, host-selection behavior, *Ophraella communa*, open field test

## Abstract

**Simple Summary:**

*Ophraella communa* is an effective biocontrol agent against the invasive common ragweed *Ambrosia artemisiifolia.* However, whether some closely related non-target plants can become alternative host plant species of *O. communa* in China remains unclear. Although extensive host-plant selection tests have been used to ensure the host specificity of *O. communa* in other countries, some doubts remain. In this study, we conducted a series of choice experiments in outdoor cages and open fields to determine the preference of *O. communa* for *A. artemisiifolia* and three non-target plant species: sunflower (*Helianthus annuus*), cocklebur (*Xanthium sibiricum*), and giant ragweed (*Ambrosia trifida*). The results showed that this beetle poses no threat to the biosafety of *H. anunuus* or *A. trifida* and exhibits a robust dispersal capacity to find and feed on *A. artemisiifolia*. However, in the future, we should be aware that *X. sibiricum* has the potential to be an alternative host plant for *O. communa*.

**Abstract:**

Understanding the host-selection behavior of herbivorous insects is important to clarify their efficacy and safety as biocontrol agents. To explore the host-plant selection of the beetle *Ophraella communa*, a natural enemy of the alien invasive common ragweed (*Ambrosia artemisiifolia*), we conducted a series of outdoor choice experiments in cages in 2010 and in open fields in 2010 and 2011 to determine the preference of *O. communa* for *A. artemisiifolia* and three non-target plant species: sunflower (*Helianthus annuus*), cocklebur (*Xanthium sibiricum*), and giant ragweed (*Ambrosia trifida*). In the outdoor cage experiment, no eggs were found on sunflowers, and *O. communa* adults rapidly moved from sunflowers to the other three plant species. Instead, adults preferred to lay eggs on *A. artemisiifolia*, followed by *X. sibiricum* and *A. trifida*, although very few eggs were observed on *A. trifida*. Observing the host-plant selection of *O. communa* in an open sunflower field, we found that *O. communa* adults always chose *A. artemisiifolia* for feeding and egg laying. Although several adults (<0.02 adults/plant) stayed on *H. annuus*, no feeding or oviposition were observed, and adults quickly transferred to *A. artemisiifolia*. In 2010 and 2011, 3 egg masses (96 eggs) were observed on sunflowers, but they failed to hatch or develop into adults. In addition, some *O. communa* adults crossed the barrier formed by *H. annuus* to feed and oviposit on *A. artemisiifolia* planted in the periphery, and persisted in patches of different densities. Additionally, only 10% of *O. communa* adults chose to feed and oviposit on the *X. sibiricum* barrier. These findings suggest that *O. communa* poses no threat to the biosafety of *H. anunuus* and *A. trifida* and exhibits a robust dispersal capacity to find and feed on *A. artemisiifolia*. However, *X. sibiricum* has the potential to be an alternative host plant for *O. communa*.

## 1. Introduction

Exploring and understanding host specificity is an important part of the methodology for selecting biological control agents [1,2,3], and the concept of insect behavior has been widely applied to improve testing for host specificity [4,5,6,7,8]. During the long-term evolutionary process, herbivorous insects have developed a series of specialized behavioral strategies to distinguish between host and non-host plants [9], and multiple mechanisms have been proposed to underlie host-plant selection [10,11,12]. For example, the “preference-performance hypothesis” predicts that female insects evolve to oviposit on hosts on which their offspring will fare best [13,14]. To maximize overall fitness, herbivorous insects must assess host-plant quality, both between and within species, and locate and select the most suitable host for feeding and larval development [15,16,17].

Although many agents used for the biological control of weeds exhibit extreme host specificity, the endogenous conditions of the insect and test arena may cause host-plant selection behavior to become more labile, thus affecting the host-plant range [18,19]. There are numerous types of host specificity tests, including choice tests [20], non-choice tests [20,21], cage tests [22], and open field choice tests [21]. The selection of the test, and even the distribution pattern of the test plants, affects the host specificity results. For example, *Microthrix inconspicuella* is a potential agent for the control of the polygonaceous weed *Emex australis*, and, under caged quarantine conditions, the larvae of this moth have been found to feed on apples, a rosaceous crop [23]. However, under field conditions or when the larvae are contained in large sleeve cages, apple foliage is not attacked [22]. Therefore, the outcomes of host specificity tests typically vary under different test designs, owing to behavioral factors.

Several behavioral factors influence test results, including sequential behavioral responses during host-plant selection [24,25,26], experience and learning [27,28,29,30,31], and time-dependent effects [32]. These factors may lead to two types of false results. A false positive result refers to an attack during the test but no potential for attack under field conditions, whereas a false negative result occurs when a plant species is not attacked during the test but might be attacked in the field. For example, if non-target plants are near the host plant, they may be more prone to attack, and insects may habituate to and accept non-host plants through repeated contact, thus leading to a false positive result. False results may lead to the rejection of potential biological control agents that might be adequately host-specific, or to the release of candidate agents that may attack non-host plants in the field.

To minimize the potential for false results, many test methods have been designed, including the use of large arenas [33], natural arenas [4], open field testing [34], and behavior-based host-selection tests, which should indicate whether a plant is susceptible to feeding or oviposition by a biological control agent under any set of field conditions [1,5]. In this study, we experimentally analyzed under field conditions the host-selection behavior of a potential biological control agent, *Ophraella communa* (Coleoptera: Chrysomelidae), against the common ragweed *Ambrosia artemisiifolia* (Asteraceae).

*Ophraella communa* has been found to be an effective agent for the biological control of common ragweed, a widespread and harmful invasive alien weed [35,36], and has achieved great success in China [37]. It is an oligophagous leaf beetle that feeds on plants of the Asteraceae family. Several studies have focused on its host range, and it has been reported to attack cockleburs (*Iva axillaris*, *Xanthium strumarium*, *X. canadense*, and *X. italicum*), giant ragweed (*Ambrosia trifida*), sunflower (*Helianthus annuus*), feverfew (*Parthenium hysterophorus*), and Jerusalem artichoke (*H. tuberosus*) (Asteraceae) [2,35,36,38]. Watanabe and Hirai, Hu and Meng, Kim et al., and Kim and Lee concluded that *O. communa* could feed on sunflower plants and even complete generations [36,38,39,40]. Therefore, this beetle was rejected for release as a biocontrol agent for ragweed because of the possible damage to crops in Australia [41]. Although extensive host-plant selection tests have been used to ensure the host specificity of *O. communa* [2,42,43,44], some doubts remain, such as whether cockleburs can become an alternative host-plant species and the host-plant range expansion of *O. communa* in China is unclear. The risk of attack by *O. communa* and the subsequent level of damage that might occur in sunflower crops under field conditions remain unknown [19,36,45].

Host and non-host plants often coexist under natural conditions, and the “physical obstruction hypothesis” describes the situation in which host plants are effectively hidden by large or tall non-host plants [13], which are usually used to protect crops from pest infestations in the field. Similarly, a biological control agent may have difficulty locating a targeted invasive host when the plant coexists with larger or taller non-host plants. In China and Europe, *A. artemisiifolia* has become a major agricultural weed, especially spring-sown crops, such as sunflower and maize [46,47,48]. Sunflower is a large and tall plant that can easily act as a barrier to biological control. Therefore, when *O. communa* is used to control *A. artemisiifolia* in sunflower cultivation, it is unclear whether the weed can hide among the crops, leading to a reduction in biological control efficiency.

Understanding the characteristics of the host-plant selection behavior of *O. communa* is important for better prediction and evaluation of its safety and efficacy as a biological control agent. It is also important to determine whether cockleburs can become alternative host-plant species in China, what is the risk of attack by *O. communa* on sunflower, and the control efficacy of *A. artemisiifolia* under field conditions. Therefore, in this study, outdoor cage and open field tests were performed to investigate the host-selection behavior of this beetle in the hope of answering these questions.

## 2. Materials and Methods

### 2.1. Host Plants

Seeds of *A. artemisiifolia* were collected from the Institute of Plant Protection of the Hunan Academy of Agricultural Sciences (IPP, HAAS, 25°21′17.81″ N, 114°33′40.00″ E), and *X. strumarium* and *A. trifida* seeds were collected from the experimental farm of Shenyang Agricultural University, Liaoning Province (41°48′ N, 123°24′ E). Seeds of *H. annuus* (oil sunflower, cv. CH609) were purchased from Ku-Fu-Tian Seed Company, Inner Mongolia Autonomous Region, China.

The *A. artemisiifolia*, *X. strumarium*, and *A. trifida* seeds were sown in individual seed trays with sterilized nutritional soil (Langfang Dingxin Seedling Company, Langfang, China) and individually transplanted into plastic pots (15 cm in diameter and 10 cm in height) with loamy clay soil at the three- to four-leaf stage. *H. annuus* seeds were sown directly in the same plastic pots. The seedlings were placed in an unheated and naturally lit greenhouse at the Langfang Experimental Station of the Institute of Plant Protection, Chinese Academy of Agricultural Sciences (LF Station, IPP, CAAS) in Langfang City, Hebei Province (39°30′42″ N, 116°36′07″ E) and watered every four days. All individuals of each species were used in the experiments when they were approximately 20–30 cm in height.

### 2.2. Insect Culture

*Ophraella communa* pupae were collected from IPP and HAAS and used to construct colonies on *A. artemisiifolia* plants at the LF Station, IPP, and CAAS. The *O. communa* population was maintained in an unheated greenhouse under a 16 h L:8 h D photoperiod at 26 ± 1 °C and 70 ± 10% relative humidity (RH) and was used for the experiments after three generations.

### 2.3. Distribution and Oviposition Preference Behavior of O. communa Adults on Four Different Coexisting Plant Species in Outdoor Cages

The experiments were conducted in outdoor cages (6.5 × 24.5 m) at LF Station, IPP, CAAS. Five sample plots (2.5 × 4.5 m) were regularly arranged in the field, and each plot was covered with a single mesh cage (2 m in height) on 2 July 2010 (Figure 1A). Two plants of each of the four tested species (approximately 30–40 cm in height) were transplanted into the above cages on 8 July 2010. The planting patterns are shown in Figure 1B. *O. communa* adults at 2 days of age were randomly collected from the greenhouse, and on 15 July 2010, 10 pairs (female: male = 1:1) were released on each sunflower plant in each cage. From the day after release to the 5th day, the numbers of adults and eggs on each plant in each cage were counted daily; the observations were performed every other day until the 27th day after release (one *O. communa* generation). Five replicates were performed for each experiment.

### 2.4. Host-Plant Selection Behavior of O. communa on Regularly Distributed Ragweed Patches in Sunflower Plots

The experiments were conducted in an open field (45 × 70 m) at the LF Station, IPP, and CAAS in 2010 and 2011. Six sample plots (20 × 20 m) were prepared, and *X. strumarium* was planted among different barrier bands (Figure 2A). The intercropping patterns of *A. artemisiifolia* and *H. annuus* are shown in Figure 2B. In each plot, 20 *A. artemisiifolia* plants were evenly planted in the center of each plot (shaded area in the figure with a 1 m radius), and 24 sunflowers were planted in a homocentric ring with a 3 m radius to create the sunflower barrier. Twenty-seven and forty-five sunflowers were planted in homocentric rings with radii of 6 and 9 m, respectively. One, two, and three *A. artemisiifolia* plants per cluster were evenly intercropped at intervals of three sunflowers in the 6 m radius homocentric ring and at intervals of four sunflowers in the 9 m homocentric ring. Sunflower and ragweed seedlings were planted 80 cm apart. *O. communa* adults at 2 days of age were randomly collected from the laboratory culture; on July 4 of both years, 40 pairs were released on *A. artemisiifolia* in the center of each plot. After three days, visual sampling was used to count the numbers of *O. communa* adults, eggs, larvae, and pupae on *A. artemisiifolia* and sunflower plants in each plot, and observations were performed every six days until the sunflower fruit ripened on September 26. Six replicates were performed for each experiment and continued for two years.

### 2.5. Data Analysis

Statistical analyses were performed using the SAS system for Windows V8. The experimental data were checked for normality and homoscedasticity, and if required, were arcsine square-root or log-transformed before analysis. In the outdoor cages experiment, three-way ANOVA followed by the Tukey test (*p* values ≤ 0.05) was performed to compare the data on *O. communa* distribution (adults and eggs) considering the effects of plant species, days after release and cages (blocks) and their interactions. In the open field experiment, preliminary analyses indicated no significant effects of year and blocks (plots). Therefore, a three-way ANOVA followed by the Tukey test (*p* values ≤ 0.05) was used to test for the effects of plant species, distance from center, and ragweed cluster density on the cumulative densities of the different *O. communa* developmental stages.

## 3. Results

### 3.1. Distribution of O. communa Adults on Four Different Coexisting Plant Species in Outdoor Cages

In the outdoor cages experiment, the results of three-way ANOVA indicated that only plant species had a significant effect on dynamics of *O. communa* adults, whereas days after release, cages and their interactions had no significant effects (Appendix A). *O. communa* adults released on *H. annuus* moved rapidly to the other three plant species, and there was a significantly lower number of *O. communa* adults on sunflower compared to the other plant species (Figure 3). After release, approximately 70% of the beetles were observed on the four different plant species, approximately 32.5% and 25% moved to *A. artemisiifolia* and *X. sibiricum*, respectively, approximately 2% moved to *A. trifida* (two adults on one plant), and approximately 10% remained on the sunflower plants. On the third day after release, most adult beetles were found to feed on *A. artemisiifolia* and *X. sibiricum.* Only one adult remained on a single sunflower plant, and several tiny feeding spots were observed; however, this area was negligible compared to the entire leaf area. After five days, no adult beetles were found on the sunflowers, but the population of adult *O. communa* remained high in *A. artemisiifolia* and *X. sibiricum*. One or two adults occasionally fed on *A. trifida*.

### 3.2. Oviposition Preference Behavior of O. communa Adults on Four Different Coexisting Plant Species in Outdoor Cages

In the outdoor cages experiment, the results of the three-way ANOVA showed significant effects of the plant, the day, and their interaction on the dynamics of *O. communa* egg deposition (Appendix A). By tracking the movement of *O. communa* adults among the four tested plant species, we found that they preferred to lay eggs on *A. artemisiifolia* followed by *X. sibiricum*. Very few eggs (<60) were observed on one *A. trifida* plant and no eggs were found on *H. annuus* plants during the entire survey period. The oviposition of *O. communa* on *A. artemisiifolia* showed a significant peak of 623.0 eggs per plant on 24 July 2010, which was significantly higher than that on *X. sibiricum* (146.2 eggs per cage) and *A. trifida* (9 eggs per cage) (Figure 4).

### 3.3. Host-Plant Selection Behavior of O. communa on Regularly Distributed Ragweed Patches in Sunflower Plots

In the open field experiment, the results of the three-way ANOVA indicated that only plant species had a significant effect on the number of *O. communa* individuals in different developmental stages, whereas distance, density, and the interactions between the three factors had no significant effects (Appendix A). In 2010 and 2011, there were significant differences in the number of *O. communa* individuals at different developmental stages in *A. artemisiifolia* compared with *H. annuus* (Figure 5). In both years, the number of *O. communa* adults on common ragweed was significantly higher than that on sunflower (Figure 5a,b). The number of eggs laid also showed consistency (Figure 5c,d). Very few eggs were found on sunflowers, and all died during development. Moreover, the number of larvae (Figure 5e,f) and pupae (Figure 5g,h) on sunflower was significantly lower and was close to zero.

In addition, based on two years of observation and records, the *O. communa* adults were mainly found feeding and/or ovipositing on *A. artemisiifolia* planted in the center of each plot during the early period after release, and very few adults were found on *A. artemisiifolia* planted in homocentric rings with radii of 6 m and 9 m (Appendix A). During the entire survey period, from July to September, the *O. communa* population completed two generations on *A. artemisiifolia* planted in the center and one generation in both homocentric rings. By September (60 days after release), almost all *A. artemisiifolia* planted in the center had died, and adult *O. communa* had moved to *A. artemisiifolia* planted in both homocentric rings to feed and oviposit (Appendix A).

## 4. Discussion

The host-plant selection behavior of herbivorous insects is complex. When larval and/or adult insects encounter target or non-target plants, the morphology and chemical properties of the plant surfaces are first evaluated by the contact receptors (antennae, mouthparts, ovipositors) of the insects, and the inner chemical characteristics of the plants are assessed to determine whether they are acceptable or antagonistic [49]. In non-choice tests, herbivorous insects are typically confined to only one test plant species; therefore, they tend to have a broader host range than in choice tests [5,20,50]. Host range overestimation may lead to the rejection of candidate biological control agents that are adequately host-specific under field conditions [4]. The risk of *O. communa* feeding on sunflower is negligible because the leaf beetle is occasionally found on *H. annuus* when all *A. artemisiifolia* plants are defoliated near the sunflower field. If the beetle feeds only on sunflower, the number of offspring will be reduced and the beetle cannot survive [42]. To date, there has been a debate on whether *O. communa* can feed on and damage *H. annuus* even though host specificity tests have been conducted for nearly 30 years.

In our field cage test, several tiny feeding spots from adult *O. communa* were found on sunflower leaves. However, those adults left the sunflower plant in the next survey (four days after release) and did not feed or oviposit on the sunflower thereafter. In the open field investigation, adult *O. communa* released on *A. artemisiifolia* in the center of the plot primarily fed and oviposited there. As *O. communa* spread to the periphery, several adults were occasionally found on sunflowers, but no feeding or oviposition behavior was observed. Our results demonstrate that adult *O. communa* are averse to sunflower compared with ragweed. “Preference-performance hypothesis”, also known as the “mother knows best” hypothesis, predicts that females prefer a host that assures the greatest fitness of their offspring [14,51,52]. In our study, no *O. communa* eggs were found on sunflowers in the cage test, but three egg masses were found on sunflowers in open sunflower fields in 2010 and 2011. However, only one egg mass hatched, and all larvae died during development. These results support the conclusion that sunflower is an unsuitable host plant for *O. communa* offspring and are consistent with findings from previous studies carried out in Canada [42] and China [43,44,53]. In addition, it is worth noting that we should be alert to the possibility of individuals dispersing from outside the field into the experimental plots in the open field experiment, because ragweed leaf beetles are known to disperse over long distances. Yamanaka et al. [54] found that, with the passage of time, *O. communa* spills over to adjacent locations at roughly the one-beetle-generation time scale according to the “resource concentration hypothesis” and “reaction–diffusion theory”. In addition, herbivorous insects can find their host plants over long distances to feed and oviposit, even though the host plants are hidden in a range of other plants and plant volatile organic compounds play an important role in the host location process. Insects rely on a powerful olfactory system, with olfactory receptor neurons able to identify volatiles cues, made by specific key compounds or specific blends emitted from suitable host plants [55,56]. For example, diterpene hydrocarbons released by the seedlings of brassicaceous hosts *Brassica oleracea* and *Brassica napus* species, alone or in combination with one or more minor compounds, are key vectors for host localization by *Bagrada hilaris* [57,58].

In many herbivorous species, female adults avoid reproduction in places where their offspring are at a high risk of predation [59,60,61,62]. In this study, many natural enemies, such as ladybeetles (*Harmonia axyridis* and *Coccinella septempunctata*), lacewings (*Chrysopa* spp.), and Pentatomidae, were observed on sunflower leaves (data not shown). This study confirmed that *O. communa* is safe for use as a biological control agent to control ragweed, based on its host-selection behavior in an open field experiment. However, when all common ragweed plants are completely eradicated or defoliated from the local population, the leaf beetle *O. communa* of suboptimal alternative host plants (*A. trifida* and *H. annuus*) should not be ignored. It has been reported that *O. communa* began feeding on *A. trifida* after all *A. artemisiifolia* plants were defoliated under field conditions in Japan [63].

In our choice test using closed cages in the field, most *O. communa* adults (25%) moved rapidly from *H. annuus* to *X. sibiricum*, in addition to the host-plant *A*. *artemisiifolia*, to feed and lay eggs, and several were found feeding and oviposition on *X. canadense* but not on *H. annuus*. Additionally, when *A. artemisiifolia* died in October, many *O. communa* adults moved to *X. canadense* to prepare for overwintering (unpublished data). In Japan, *O. communa* can also be found feeding on *X. canadense*, and adults in the field have been found to move to *X. canadense* overwinter after the death of *A. artemisiifolia* in late summer [39]. Our results indicated that *X. sibiricum* may be a suitable host-plant species for this beetle. This result is consistent with those of Cao et al. [43] and Liu et al. [64], who suggested that *X. sibiricum* could be used as a lower-ranked host plant next to the target weed. In China, *X. canadense* is a common weed in cultivated fields, especially in soybean, tobacco, and sunflower fields [65,66,67]. Therefore, *O. communa* may be used to control *X. canadense* in China in the future.

In its native range, *O. communa* does not utilize *A. trifida* as a host plant [36,54,68,69], and, in our study, although *A. trifida* was attacked by several adults, the damage level was very low. Therefore, the beetle cannot effectively control *A. trifida*, and there have been no reports on the use of *O. communa* to effectively control *A. trifida* in China. However, this beetle has been reported to feed extensively on *A. trifida* in fields throughout the Japanese islands [35,36,63]. These results indicate the expansion of the host range of occasionally introduced *O. communa*, which may be the result of the co-evolution of herbivorous insects and host plants [19,69,70].

In our study, after the adult *O. communa* that were released on *A. artemisiifolia* plants in the center of the sunflower field completed one generation (approximately 30 days), they initiated a search for suitable host plants across the sunflower barrier (planted in homocentric rings with a radius of 3 m), and the number of *O. communa* in the peripheral population was supplemented by the population in the center. This result indicated that *O. communa* has a robust capacity to find *A. artemisiifolia* for feeding. The “resource concentration hypothesis” predicts that specialist herbivorous insects are more likely to find and stay longer on host plants growing in dense or nearly pure contexts [71,72,73]. In our study, the cumulative densities of *O. communa* feeding or remaining on *A. artemisiifolia* did not differ among plant clusters of different densities, which does not support the “resource concentration hypothesis”. This result was consistent with the observations of Yamanaka et al. [54]. Insects with high dispersal abilities may not be limited by patch borders. Hence, their densities per plant did not differ among host-plant patches of different sizes [74]. In addition, if patches are closer together, insects may move more easily between them, thus diminishing differences in density [75,76]. *O. communa* has been shown to rapidly disperse after introduction into a new area [77,78,79], and our results further support this high dispersal ability. Certainly, the proximity of the *A. artemisiifolia* clusters (< 4.5 m) may have resulted in a lack of difference in the densities of *O. communa* in plants. In this study, we used well-established plants as test plants (all 135 individuals of each species were used for the experiments when they were approximately 20–30 cm in height). However, sunflower was sown; thus, seeding and younger sunflower plants were exposed to *O. communa* and these might be more susceptible to feeding and oviposition. More experiments are needed to confirm this in the future. Finally, how can we better predict the long-term benefits and risks of ragweed biology control? We advocate research on host specificity and population differentiation before the release of biocontrol agents to promote the development of improved biological control under changing global conditions [3,19,80].

## 5. Conclusions

In summary, by observing the host-plant selection behavior of *O. communa*, we conclude that this beetle poses no threat to the biosafety of *H. annuus*. In addition, *X. sibiricum* has the potential to become an alternative host plant for *O. communa* in the future; however, it cannot efficiently control *A. trifida* in China. In the open field study, some *O. communa* adults crossed the barrier formed by *H. annuus* to feed and lay eggs on *A. artemisiifolia* planted in the periphery, and the spatial interactions between *A. artemisiifolia* and *O. communa* did not support the “resource concentration hypothesis”. We conclude that *O. communa* has a robust dispersal capacity to find and feed on *A. artemisiifolia*.

## Figures and Tables

**Figure 1 insects-14-00334-f001:**
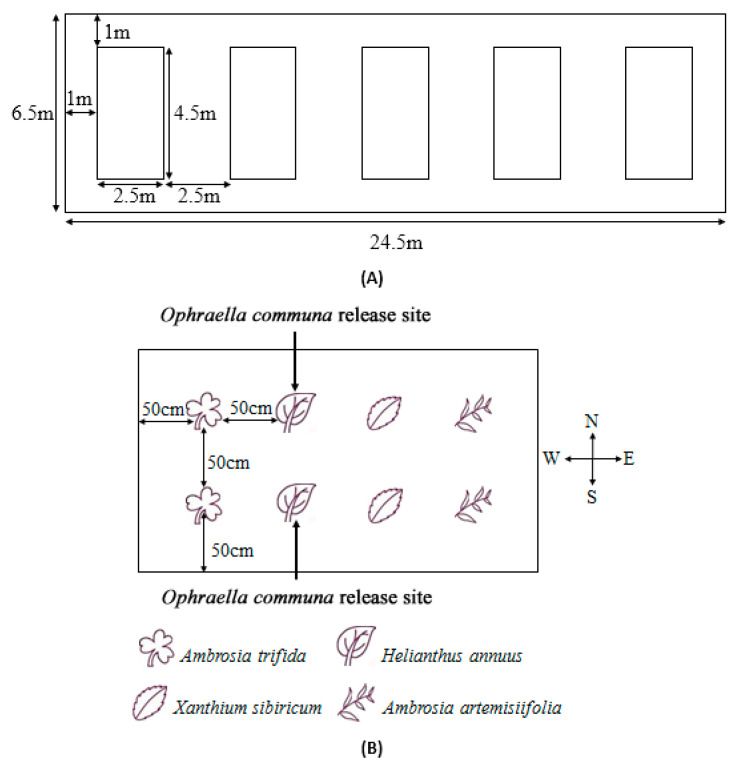
(**A**) Arrangement of experimental plots, (**B**) planting pattern, and *Ophraella communa* release sites in each plot.

**Figure 2 insects-14-00334-f002:**
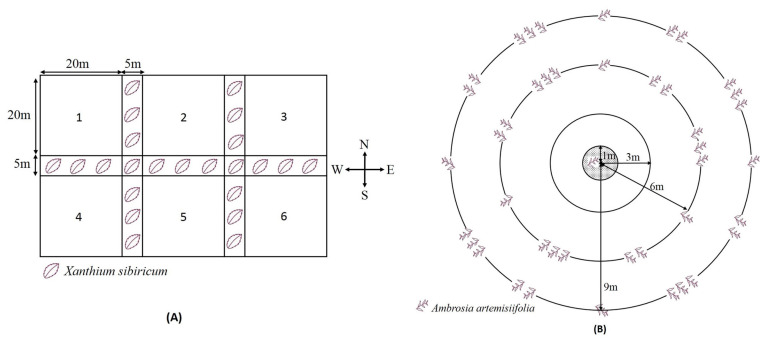
(**A**) Arrangement of experimental plots and (**B**) the planting pattern for the *Ophraella communa* host-plant selection behavior test on regularly distributed ragweed patches in sunflower plots. The shaded area indicates that 20 ragweed plants were evenly planted in the center (a radius of 1 m) of each plot. “
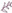
” shows the sites and densities per cluster of common ragweed planted in the 6 m and 9 m radius homocentric rings in each plot.

**Figure 3 insects-14-00334-f003:**
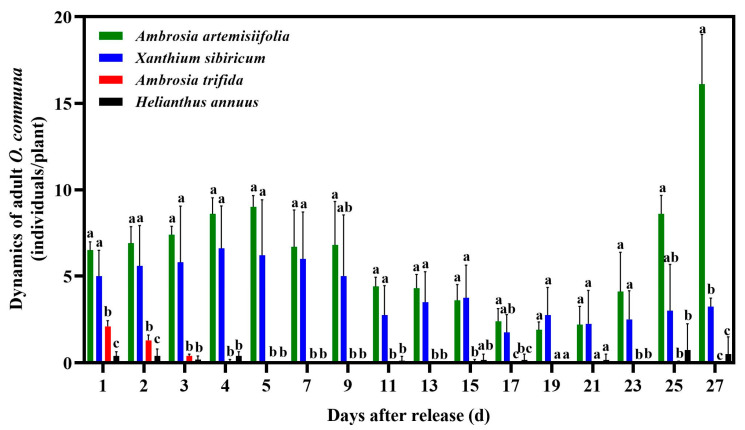
Dynamics of adult *Ophraella communa* occurrence on four coexisting plant species in cage experiments. All values are shown as the means ± SE. Values with the same letter in the same day are not significantly different (three-way ANOVA followed by Tukey test at *p* ≤ 0.05).

**Figure 4 insects-14-00334-f004:**
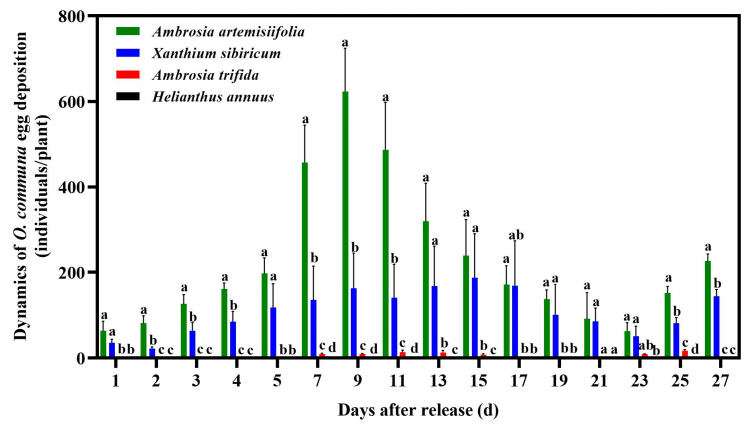
Dynamics of *Ophraella communa* egg deposition on four coexisting plant species in cage experiments. All values are shown as the means ± SE. Values with the same letter in the same day are not significantly different (three-way ANOVA followed by Tukey test at *p* ≤ 0.05).

**Figure 5 insects-14-00334-f005:**
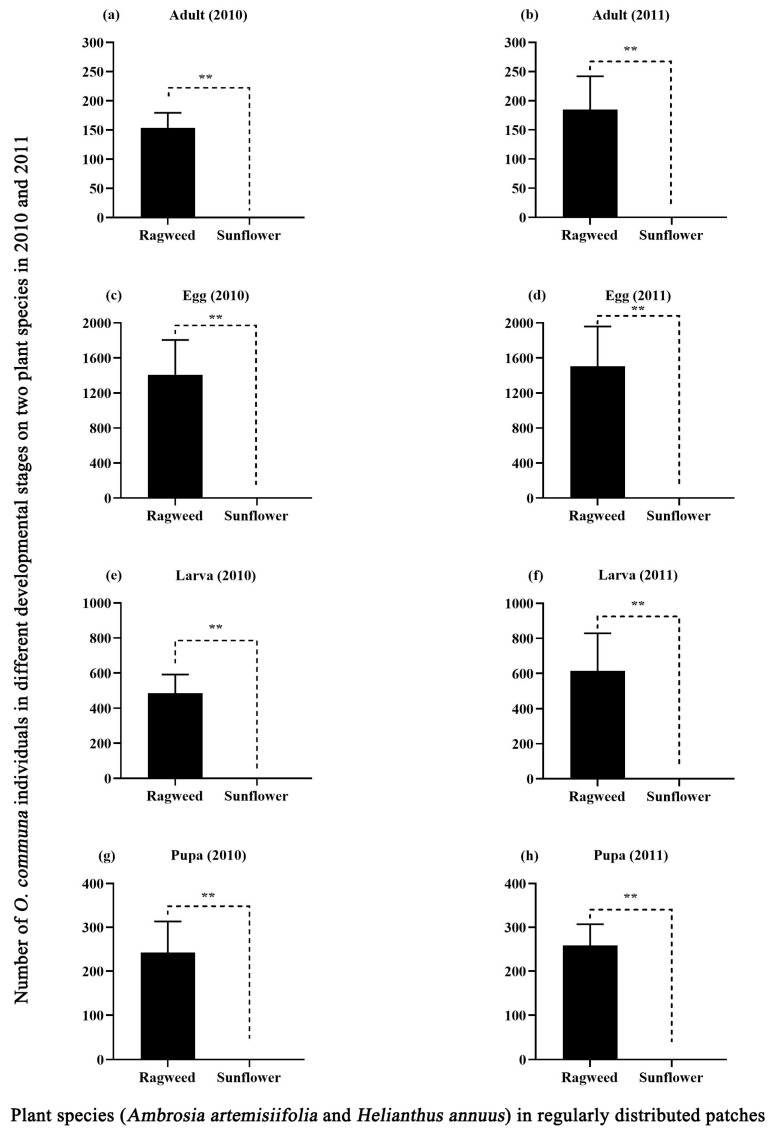
The number of *O. communa* individuals in different developmental stages on *A. artemisiifolia* (individuals/plant) and *H. annuus* (individuals/plant) planted in 2010 and 2011. All values are shown as the means ± SE. ** *p* < 0.01, highly significant. (**a**): the number of O. communa adults on *A. artemisiifolia* and *H. annuus* in 2010, (**b**): the number of *O. communa* adults on *A. artemisiifolia* and *H. annuus* in 2011,(**c**): the number of *O. communa* eggs on *A. artemisiifolia* and *H. annuus* in 2010., (**d**): the number of *O. communa* eggs on *A. artemisiifolia* and *H. annuus* in 2011.,(**e**): the number of *O. communa* larvae on *A. artemisiifolia* and *H. annuus* in 2010.,(**f**): the number of *O. communa* larvae on *A. artemisiifolia* and *H. annuus* in 2011.,(**g**): the number of *O. communa* pupae on *A. artemisiifolia* and *H. annuus* in 2010, (**h**): the number of *O. communa* pupae on *A. artemisiifolia* and *H. annuus* in 2011.

## Data Availability

Data used in this study are available from the corresponding authors upon reasonable request.

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
