# Peer review of "Host-Plant Selection Behavior of Ophraella communa, a Biocontrol Agent of the Invasive Common Ragweed Ambrosia artemisiifolia"

_insects, 2023, doi:10.3390/insects14040334_

Round 1

Author Response

Dear editor and reviewer:

Major revisions of the manuscript were completed and our manuscript has been carefully edited and proofread by professional editing organizations. We would like to thank Editage (www.editage.cn) for English language editing. All these comments from editor and reviewers were addressed carefully. Now the manuscript was resubmitted on line. Thanks for your work and valuable comments.We deeply appreciate your consideration of our manuscript, and we look forward to receiving comments from the reviewers. If you have any queries, please contact us.

Sincerely yours

Reviewer 2 Report

The paper is reporting experiments on host-plant selection behaviour of Ophraella communa, a biocontrol agent of an invasive weed. The topic of the paper has importance. The type of experimental approach, focused on semi-field and field experiment is nice; English style is fair, but few parts of the paper in introduction and discussion are fragmentary. Overall, several aspects require strong attention across the manuscript. Materials and methods section sometimes miss of important information. In particular, the materials and methods and the results of the second experiments are pretty confusing. This makes difficult to replicate the experiment. The materials and methods paragraphs and the results paragraphs should coincide. In figures there is no show of the statistical values, so is not easy to understand in brief which are the main results obtained. In discussion the role of semiochemicals in determining herbivore-host plant finding process is overlooked.

Line 20, rewrite this concept.

Line 23-26, the conclusion doesn’t seems supported by the results.

Line 65-66, give some references.

Line 92, don’t abbreviate scientific name at the beginning of a sentence.

In introduction, why to stress so much that laboratory experiment can give false results? I agree that the results in laboratory without field work are incomplete/useless, but laboratory bioassay are the basic first step. In general, I think introduction should be better re-organized, pointing out better the objectives in the end.

line 145, I understand choice and oviposition estimation, but how was feeding estimated? Is not reported. Other details missing, how many times was the experiment replicated? In addition, I understood that the experiment was lasting 5 days, while in results the observations seem carried pout for 26 days. Please advise.

Line 158, despite the authors ‘efforts, still the experiment is not very clear. What is the hypothesis that authors want to demonstrate?

Line 175, define better the time of the experiment.

Line 207-208, how was feeding estimated?

Figure 3 and 4 are not showing statistical differences. The resolution of the figures is rather poor.

Lines 234-235, 240, the concepts of “high” and “low” are very relative and should be avoided in results.

Line 234-248 this part is rather confusing and I cannot see any real statist value that is evidencing some clear result.

Figure 5 again is not showing any statistic; I suggest to make an ANOVA followed by Tukey or Bonferroni test to evidence with different letters indicating significant statistical differences among means.

Line 278-284, difficult to follow, maybe a table would make it clearer.

Line 312-317, I think these concepts should be pointed out in introduction.

Lines 413-414, avoid to resume here what was done, just focus on the main achievement and future perspectives.

I understand this is not the main focus of the paper, but the discussion is totally avoiding the role of plant VOC in the herbivore host selection see for example:

Bruce, T. J., & Pickett, J. A. (2011). Perception of plant volatile blends by herbivorous insects–finding the right mix. Phytochemistry72(13), 1605-1611.

Bruce, T. J., Wadhams, L. J., & Woodcock, C. M. (2005). Insect host location: a volatile situation. Trends in plant science10(6), 269-274.

Guarino, S., Arif, M. A., Millar, J. G., Colazza, S., & Peri, E. (2018). Volatile unsaturated hydrocarbons emitted by seedlings of Brassica species provide host location cues to Bagrada hilaris. PloS one13(12), e0209870.

Arif, M. A., Guarino, S., Peri, E., & Colazza, S. (2020). Evaluation of Brassicaceae Seedlings as Trap Plants for Bagrada Hilaris Burmeister in Caper Bush Cultivations. Sustainability12(16), 6361.

Author Response

(The authors gave the same response as above.)

Round 2

Reviewer 2 Report

Overall the paper has been significantly improved.

The titles of the paragraphs in materials and methods and results are different and should coincide to make it easier to read.

Some more comment:

In table 2 there is not data that is statistically significant. There are several possible interactions among factors: year * density, year * distance, distance * density. Why not consider all the interaction? Overall the table indicate that years, distance, and density don’t influence the cumulative number of individuals, is it correct? In my opinion, in consideration of this, probably the table could go in supplementary material section and the main results of this statistic summarized in the result section.

In addition, line 306-309, instead of reporting details that should go in material and method section the authors should report that this a mean comparison with factors “year”, “density” and “distance” that might have or not influenced the number of individuals. So the sentence can be deleted.

Line 95-96, I would change in “Several studies report” putting all the references at the end of the sentence.

Figure 6 need to be revised as hard to read, the characters are too small.

Line 356, revise as “Insects rely on a powerful olfactory system, with olfactory receptor neurons able to identify volatiles cues, made by specific key compounds or specific blends emitted from suitable host plants”

Author Response

Dear editor and reviewer:

Minor revisions of the manuscript were completed. All these comments from editor and reviewers were addressed carefully. Now the manuscript was resubmitted on line. Thanks for your work and valuable comments.

We deeply appreciate your consideration of our manuscript, and we look forward to receiving comments from the reviewers. If you have any queries, please contact us.

Sincerely yours,

jisu Jin
